# 3D Biocomposites Comprising Marine Collagen and Silica-Based Materials Inspired on the Composition of Marine Sponge Skeletons Envisaging Bone Tissue Regeneration

**DOI:** 10.3390/md20110718

**Published:** 2022-11-16

**Authors:** Eva Martins, Gabriela S. Diogo, Ricardo Pires, Rui L. Reis, Tiago H. Silva

**Affiliations:** 13B’s Research Group, I3Bs—Research Institute on Biomaterials, Biodegradables and Biomimetics, University of Minho, Headquarters of the European Institute of Excellence on Tissue Engineering and Regenerative Medicine, AvePark, Parque de Ciência e Tecnologia, Zona Industrial da Gandra, Barco, 4805-017 Guimarães, Portugal; 2ICVS/3B’s–PT Government Associate Laboratory, Braga, 4710-057 Guimarães, Portugal

**Keywords:** marine by-products, marine collagen, biosilica, 3D composites scaffold, marine biomaterials

## Abstract

Ocean resources are a priceless repository of unique species and bioactive compounds with denouement properties that can be used in the fabrication of advanced biomaterials as new templates for supporting the cell culture envisaging tissue engineering approaches. The collagen of marine origin can be sustainably isolated from the underrated fish processing industry by-products, while silica and related materials can be found in the spicules of marine sponges and diatoms frustules. Aiming to address the potential of biomaterials composed from marine collagen and silica-based materials in the context of bone regeneration, four different 3D porous structure formulations (COL, COL:BG, COL:D.E, and COL:BS) were fabricated by freeze-drying. The skins of Atlantic cod (*Gadus morhua*) were used as raw materials for the collagen (COL) isolation, which was successfully characterized by SDS-PAGE, FTIR, CD, and amino acid analyses, and identified as a type I collagen, produced with a 1.5% yield and a preserved characteristic triple helix conformation. Bioactive glass 45S5 bioglass^®^ (BG), diatomaceous earth (D.E.) powder, and biosilica (BS) isolated from the *Axinella infundibuliformis* sponge were chosen as silica-based materials, which were obtained as microparticles and characterized by distinct morphological features. The biomaterials revealed microporous structures, showing a porosity higher than 85%, a mean pore size range of 138–315 μm depending on their composition, with 70% interconnectivity which can be favorable for cell migration and ensure the needed nutrient supply. In vitro, biological assays were conducted by culturing L929 fibroblast-like cells, which confirmed not only the non-toxic nature of the developed biomaterials but also their capability to support cell adhesion and proliferation, particularly the COL:BS biomaterials, as observed by calcein-AM staining upon seven days of culture. Moreover, phalloidin and DAPI staining revealed well-spread cells, populating the entire construct. This study established marine collagen/silica biocomposites as potential scaffolds for tissue engineering, setting the basis for future studies, particularly envisaging the regeneration of non-load-bearing bone tissues.

## 1. Introduction

Natural bone contains the bone extracellular matrix (ECM), cells, and bioactive factors, with ECM being composed of a mixture of inorganic minerals and organic polymers [1]. In more detail, this inorganic–organic biocomposite is formed by approximately 70 wt% inorganic materials of calcium phosphates (CaP), including hydroxyapatite (HAP), 10 wt% of water, and 20 wt% of an organic matrix mainly constituted by collagens and others proteins, such as proteoglycans and glycoproteins, with a hierarchical structural organization [2]. The bone is a dynamic tissue that, alongside human life, is constantly remodeling, but in situations of bone fracture or large defects caused by disease, these mechanisms are disturbed, resulting in serious problems. The current therapeutic procedures may include the use of autografts (transferring the bone of the own patient to another location), allografts (the use of cadaver bone), alloplastic (bone-forming scaffold building with natural or synthetic materials) or xenografts (materials from other species). However, these approaches have certain limitations and do not represent an ideal solution as they do not promote bone regeneration.

Tissue engineering is an emerging biotechnology field that is searching for strategies to promote the generation of new tissues to fully substitute losses caused by trauma or disease, combining materials and cell approaches to manufacture tissue constructs that better mimic the targeted microenvironment. These advanced therapeutic products aim for the regeneration of damaged tissues and the full recovery of function mechanisms [3], requiring the use of biocompatible, absorbable, biofunctional materials organized as a scaffold structure, guiding cell attachment and proliferation while further enabling its replacement by the natural extracellular matrix (ECM). When targeting bone regeneration, these biomaterials should induce vascularization, osteogenesis, and osteoconduction, but present weak antigenicity [4,5]. For this reason, the demand for new materials is central when designing a blueprint for bone tissue engineering, with bioactive glasses and silica-based materials appearing as promising osteoinductive materials [6]; collagens, as the main component of the extracellular matrix, are one of the gold structural biopolymers considered for scaffolding [7].

The 45S5 Bioglass^®^ is a commercial bioactive glass material widely used in biomedical research, which has shown an important role in the enhancement of new bone formations due to its bioactivity and osteoconduction with the stimulation of calcium phosphates deposition on its surface and the ability to form an interfacial bond between the bone and the biomaterial [8]. Moreover, it has been demonstrated that silica is required for the proper development of bone connective tissue in mammals [9,10], and silica nanoparticles have been reported to induce osteogenic differentiation in hMSCs [11,12]. Diatomaceous earth (diatoms or D.E) is an important biogenic silica source resulting from deposits of natural fossilized skeleton made up of amorphous silica called a frustule and photosynthetic microalgae [13]. These materials are porous structures with a high surface area, as well as nanoscopic pores with a high application in the fabrication of materials [14] as drug delivery carriers [15], nanoparticles [16], or the development of artificial matrices [17]. Few studies have been conducted using silica isolated from marine organisms such as sponges; however, the sponge skeletons have been reported to be a remarkable template for tissue engineering approaches [18,19].

On the other hand, collagen is a key protein used for biomaterial manufacturing in the biomedical field. Collagen is widely abundant in all vertebrates and a prevalent component of the extracellular matrix (ECM), conferring mechanical structure, strength, biological integrity, and maintaining homeostasis in almost all tissues [20,21]. More than 28 different types of collagen have been identified, but type I collagen is the most abundant in the connective tissues, such as bone, skin, and cornea [22,23]. Collagen has advantageous biological and physicochemical properties, namely, excellent biocompatibility, biodegradability, and weak antigenicity [24]. Additionally, collagen substrates are being currently used as a template for cell growth and play a role in enhancing the mineralization of bone substitutes [1,25]. However, the processability of the collagen is limited by temperature range due to its susceptibility to enzymatic degradation and synthesized structures which normally show low biomechanical stiffness [26]. The characteristics of collagen-based materials are dependent on the collagen source, the method of extraction, purification, fibril formation, and the subsequent use of crosslinking agents [27,28]. Carbodiimide and N-hydroxysuccinimide (NHS) are two broad standards of collagen crosslinking reagents that have good biocompatibility, resulting in collagen-based materials with increased stiffness [29]. Moreover, the ionic crosslinking reaction may be promoted through the combination of collagens with another biopolymer, such as chitosan, in which the bond between the positively charged amino groups with negatively charged carboxyl groups of collagens is well-established [30]. Presently, the valorization of the protein-rich marine by-products is being studied by many research groups and has already been addressed by the industry, retrieving collagens from fish skins [31,32], swim bladders [33], or cartilages [34]. An alternative to the well-established mammal collagens is thus being built, with applications already established in several industrial sectors, such as food and cosmetics, while also being studied for biomedicine [35].

Silicate-based ceramics have been combined with different polymers to create versatile composite scaffolds. These materials have enhanced biological performances compared to phosphate-based ceramics [36]. Previous studies have reported the use of mammal collagens and silica materials in the design of biomaterials for tissue engineering, with collagen reported to favor the attachment and survival of the cells [37,38] and the covalent linkage of silica to polymers enabling an improvement in the mechanical properties [39,40]. Moreover, collagen-silica (CS) composite scaffolds demonstrated an improved water uptake capacity while also enhancing biological stability in certain silica concentrations (silica:collagen ratio of ≤1) [41]. 

Recently, collagen/bioactive glass nanoparticle coatings have demonstrated enhanced osteogenic differentiation in human mesenchymal stem cells [42]. Moreover, porous bioactive glass micro/nanospheres have emerged as attractive biomaterials in various biomedical applications, including bone tissue regeneration, wound healing, therapeutic agent delivery, and cancer therapy [43]. Interestingly, doxycycline (DOXY)-loaded diatom biosilica (DBs) were developed and coated with a hydroxybutyl chitosan (HBC) hydrogel, demonstrating wound re-epithelialization and accelerating the healing mechanism. Additionally, diatom biosilica showed interesting properties in the improvement of hemostasis for wound healing applications [44]. In vivo studies have demonstrated that silica is biocompatible in mice and rats [45].

One of the biggest challenges in tissue engineering is the creation of hierarchical porous scaffolds that mimic the composition and characteristics of human natural bone.

This present work aimed to obtain and characterize marine collagen (COL) extracted from codfish skins and developed tridimensional-based porous composites by its combination with silica-derived materials (bioglass, diatomaceous earth, and biosilica isolated from marine sponges) as scaffolds for bone tissue engineering. This study could evidence the use of silica materials in collagen matrixes to improve mechanical and biological properties for biomedical purposes.

## 2. Results and Discussion

### 2.1. Physicochemical Properties of the Codfish Skin Collagen

Marine collagen can be extracted from fish byproducts, such as scales, fins, and skins. However, the properties of collagen are determined by the source of the raw materials used. In this present study, collagen was successfully extracted from the skins of the *Gadus morhua* codfish species and was purified by salt precipitation and dialysis (Figure 1A). The collagen extraction was performed using an acetic acid solution, with a method that can be easily applied in an industrial context. The extracted material was obtained with a yield of 1.5% in relation to the initial wet weight of codfish skins, similar to that obtained with collagen extractions from the skin of bigeye snapper (1.59%) [46] and from the skin of pharaoh cuttlefish (1.66%) [47].

SDS-PAGE enables the estimation of the protein’s molecular weight, and the gel depicted in Figure 1B shows the subunit bands of the marine collagen isolated from the codfish skins. This result is compatible with the type I collagen as the characteristic fingerprint was observed, formed by two alpha chains (α_1_ and α_2_) at about 100 kDa, with the relative amount of α_2_ being lower than α_1_ and the additional high molecular weight component β (dimers) at around 250 kDa. These findings were in accordance with our previous work [48] and a clear resemblance of the standard of the bovine collagen type I. By its turn the FTIR spectrum exhibited the presence of characteristic peaks of amides A, B, I, II, and III (Figure 1C). Both amides A and B are indicative of the native collagen structure (triple helix), whereas amide III is considered the collagen fingerprint due to the presence of the typical collagen repeating tripeptide (Gly-X-Y) where Gly represents glycine and X or Y is any other amino acid residue, but the most common ones are proline (Pro) and hydroxyproline (OHPro), respectively [49,50]. The presence of amide A is due to the stretching vibration of N-H at 3325 cm^−1^, and amide B has a strong intensity within the range of 3075–2851 cm^−1^ due to CH stretching vibrations. The absorption band of amide I at 1649 cm^−1^, amide II at 1553/1484 cm^−1^, and amide III at 1383/1236 cm^−1^ was attributed to the stretching of NH bending associated with CN stretching and C-O stretching vibration modes, respectively. Our results are in consonance with the literature in which, namely, the spectra are disclosed for collagens obtained from other marine species [51,52,53].

In Figure 1D, the content of amino acids shows the typical composition of collagen, characterized by polypeptides within the triple-helical region consisting of the repeated sequence Gly-X-Y. The ratio of glycine (Gly), proline (Pro), and hydroxyproline (OHPro) contribute to 48.3% of the total amino acids in the collagen. Glycine has the highest content, with nearly 34.4% of the total amino acids, representing almost one-third of the protein in accordance with the triplet model indicated above and a degree of hydroxylation of 64%. The pyrrolidine amino acids (proline and hydroxyproline) content was 139/1000 for the codfish collagen, followed by alanine with 12.1%, showing how these are the four most abundant amino acids in collagens [54]. Furthermore, the hydroxyproline is exclusive of the collagens and is used to indicate its presence when analyzing protein extracts, and has usually an important role in the stability and integrity of the collagen protein, namely regarding the stabilization of the triple-stranded collagen helix [23,55,56]. According to previous findings, collagens of a marine origin show a lower denaturation temperature compared to mammalian collagens as a consequence of their low pyrrolidine amino acid contents [57]. Given this lower thermal stability, the produced collagen was evaluated by circular dichroism (CD) to assess the preservation of the native triple helix. The CD spectrum confirmed the typical collagen conformation (Figure 1E), with a random coil conformation characterized by a negative peak between 180 and 210 nm and the presence of a triple helix indicated by a positive peak at 220 nm, confirming a good quality of the isolated protein. Besides its association with thermal stability, the triple helix feature of collagen has been reported to enhance the mechanical strength of the collagen and the capability to interact with other biomolecules [55].

### 2.2. Characterization of Silica-Based Materials 

The silica-based materials used in this work, namely bioglass (BG), diatomaceous earth (D.E.), and biosilica (BS), were retrieved from *Axinella infundibuliformis* marine sponge, corresponding to Figure 2A–C, respectively, and were characterized by SEM for the assessment of the surface morphology of the materials, by FTIR for the identification of typical peaks due to the presence of silicon associated specific bonds, and by EDS for the determination of elements present in the silica-based raw-materials.

SEM micrographs of the three silica-based materials show particulate materials exhibiting different morphologies: while bioglass was characterized by uniform particles of around a 5 μm width (Figure 2D), ivory-colored diatomaceous earth displayed hierarchical structures characterizing the diatom species, with tube-like porous structures, a width of about 15 μm, and a length of about 25 μm (Figure 2E), and, by its turn, light-grey biosilica was comprised of needle-like particles—typical of a type of sponge spicules—about 5 μm in width and length ranging from a few hundred micrometers (Figure 2F). In relation to the FTIR spectra, the main absorption bands were attributed to Si-O-Si and Si-O stretching vibration modes, observed at 1000 and 930 cm^−1^ for BG (Figure 2G), at 1082.5 and 800 cm^−1^ for D.E. (Figure 2H), and at 1040 cm^−1^ and 797.5 cm^−1^ for BS (Figure 2I), characteristic of amorphous silica glass. Moreover, other significant peaks were observed for BG at 1445 cm^−1^, corresponding to the different P = vibrations, and at 745 and 610 cm^−1^ relating to the O-P-O vibration modes due to the presence of phosphorus pentoxide. Moreover, the elemental composition determined by EDS revealed the presence of oxygen and silicon, composing of silica, in all the samples. In addition, carbon was also detected, but due to the carbon tape being used to fix the materials, it was thus an experimental artifact [58]. On an individual and more detailed analysis, the elements (w%) identified in BG were oxygen (40 ± 0.3), carbon (30.5 ± 0.3), silicon (10.8 ± 0.1), calcium (10.1 ± 0.1), sodium (7.3 ± 0.1), and phosphorus (1.3 ± 0.1) (Figure 2J), in accordance to its known formulation. Regarding D.E., the identified elements were oxygen (44.9 ± 0.3), carbon (34.2 ± 0.4), silicon (19.3 ± 0.2), aluminum (1.0), iron (0.4 ± 0.1), and calcium (0.1) (Figure 2J), thus showing that the presence of small quantities of metals is largely abundant in natural minerals. Furthermore, the elements identified in BS were oxygen (44.7 ± 0.5), silicon (27 ± 0.3), carbon (24.6 ± 0.8), calcium (0.9 ± 0.1), sodium (0.8 ± 0.1), potassium (0.5), magnesium (0.4%), iron (0.3 ± 0.1), sulfur (0.2), and aluminum (0.2) (Figure 2J), again showing that the presence of small quantities of metals are abundant in natural minerals with a higher diversity than the one observed in D.E. 

### 2.3. Morphological Characterization of Collagen/Silica-Based Scaffolds

Biomaterials composed of collagen blended with silica-based materials were manufactured by lyophilization, aiming to obtain porous, interconnected structures resembling the architectural features of marine sponges and of the targeted trabecular bone tissue [19]. Four different formulations were tested: one condition only with the marine collagen (COL) and the three other conditions combining marine collagen with bioglass (COL:BG), diatomaceous earth (COL:D.E.), and biosilica (COL:BS). Moreover, the EDC crosslinker agent was used for the fabrication of cohesive 3D structures. The obtained structures are illustrated by the images depicted in Figure 3I and were obtained using a stereomicroscope (ZEISS). The morphological features of the developed scaffolds were assessed by SEM, with the obtained microphotographs showing the presence of a porous structure, with visible differences in the pore size between different formulations. Moreover, the SEM micrographs that were obtained with a higher magnification revealed the composite nature of the scaffolds, evidencing the presence of BG, D.E., and BS powders embedded in the collagen matrix (Figure 3II).

Additionally, the microcomputed tomography (micro-CT) technique allowed the three-dimensional (3-D) reconstruction of the scaffold structure with the determination of morphometric features, namely, porosity (%), pore size (µm), interconnectivity (%), pore wall thickness (µm), and the degree of anisotropy (DA) (Figure 4). The obtained images are illustrative of the different scaffold formulations showing the porous collagen matrix (green color in the 3D reconstructions—Figure 4I), with the ones regarding the composite structures (COL:BG, COL:D.E., and COL:BS) displaying the presence of denser particles (red color in Figure 4I) as well, due to the presence of the silica-based materials. The morphometric characterization demonstrated that the total porosity was higher than 85% for all the tested conditions (Figure 4II), similar to that of the human native trabecular bone, which has a porosity ranging from 50% to 90%. The lowest value was obtained for COL:BG (85 ± 2.00), and the highest value was obtained for the COL:BS formulation (96 ± 1.00). The statistical analysis revealed significant differences between these two conditions.

In general, the scaffolds’ porosity has been shown to develop a crucial role in cell behavior and distribution, but other structural parameters are also important, such as pore size and interconnectivity. From the data analysis, the pore size of COL, COL:BG, COL:D.E., and COL:BS was 315 µm ± 103.00, 138 µm ± 15.00, 244 µm ± 124.00, and 281 µm ± 12.00, respectively (Figure 4IIB), with the introduction of silica-based materials apparently leading to a decrease in the pore size, although no statistically significant differences were detected. These values are within the range of pore sizes considered favorable for cell proliferation and for new extracellular matrix formation in bone tissue engineering between 100 and 500 µm [59]. Past studies have demonstrated that the efficiency of cell attachment, migration, and infiltration is commonly affected by the structures’ pore size, with an overall assessment leading to the indicated pore size range. Smaller pores (ø < 100 µm) have been characterized by their higher surface area, promoting cell adhesion, but in contrast, they are not useful for cell penetration. Pores larger than a 100 µm diameter are preferred for their cell penetration to the central parts of the 3D structures, thus allowing the desired full population of the scaffold for which the interconnectivity between the pores is also required. In this work, the interconnectivity of the developed scaffolds was higher than 70% for all conditions (Figure 4IIC), which is considered a crucial property feature of biomaterials since it directly influences cellular communication, enabling favorable mass transport and nutrient diffusion. The processing methodology was efficient for creating structures with large and interconnected pores with no particular orientation, being thus anisotropic random structures (with an anisotropy degree less than 0.5, Figure 4IID), of the trabecular bone, with a pore wall thickness or trabeculae ranging between 20 µm and 30 µm (Figure 4IIE). However, in general, no statistically significant differences were observed between the structures of the fabricated biomaterials.

### 2.4. Elemental Analysis

Elemental analysis was performed by an energy-dispersive X-ray spectroscopy (EDS) analysis. On the outer layer/surface of the biomaterials, different chemical elements were distinguished according to the percentage relative to weight. The percentage of the chemical composition of the COL biomaterial presented with the higher content of carbon and oxygen with 58.8 and 25%, respectively. Moreover, the COL biomaterial showed a lower content of nitrogen compared with the other biomaterial conditions. The silicon element represents 0.8%, 4.7%, and 4.4% of the global composition of COL:BG, COL:D.E., and COL:BS (Table 1).

The higher percentage in weight of organic elements, namely, carbon, oxygen, and nitrogen, are present in all the produced biomaterials. These elements are characteristics of the amide functional groups (amide A, amide I), which were present on the organic collagen protein identified by FTIR analysis (Figure 1C) and also for the amino acids content (Figure 1D). This analysis allowed the confirmation of the incorporation of silica materials (the presence of silicon in these silica-derived formulations) in the collagen matrix. This presence of silicon is expected to have a positive impact on bone tissue engineering. In fact, silicon has been reported to decrease bone resorption and nucleate the precipitation of hydroxyapatite [60]. Moreover, recently, biosilica has received attention as a potential osteoinductive additive [61].

### 2.5. Mechanical Properties

In the literature, it has been reported that biomaterials produced only with collagen have limited application in tissue replacement due to poor mechanical properties and rapid degradation [62]. One strategy that is being explored to overcome this bottleneck is the use of crosslinking agents and the formulation of collagen with other components, namely, silica-based materials. The mechanical properties of the developed COL; COL:BG, COL:D.E., and COL:BS biomaterials were tested by the compression mode using INSTRON 5540, and the results are presented in Figure 5A. The addition of silica-based materials enabled a light reinforcement of the compressive modulus from 1.02 ± 0.20 kPa of the bare collagen structure to 2.2 ± 0.29 kPa (COL:BG), 1.56 ± 0.51 kPa (COL:D.E) or 1.22 ± 0.08 kPa (COL:BS), although only the former showed significant statistical differences (*p* < 0.05) when compared to COL. These observations are in agreement with the literature, where it has been reported that the incorporation of inorganic components could reinforce collagen scaffolds, namely, with a compressive modulus of up to 1 MPa [63,64]. Considering the reference values of compressive modulus, the developed 3D biomaterials showed an order of magnitude lower than that of trabecular (10.4 GPa ± 3.5) and compact bone (18.6 GPa ± 3.5) [65,66]. Still, due to their predominantly biocompatible nature, and despite low mechanical properties, different absorbable collagen sponges are used in clinics, exhibiting values of compressive modulus between 1 and 20 kPa, in the same range as the ones obtained in the present study [67,68].

### 2.6. Absorption Capacity

The swelling and hydrophilic property of the developed 3D biomaterials was evaluated as their water absorption capacity, with the obtained results shown in Figure 5B. The produced biomaterials had shown a superabsorbent capacity due to the network of interconnected pores and a rapid swelling capacity in less than 60 min. It was shown that the absorption ratio of the COL biomaterials was higher (1905%) when compared to the other biomaterials (931%, 754%, and 798% for COL:BG, COL:D.E., and COL:BS, respectively), even after immersion for only 1 min. In addition, after 5 min of incubation in PBS, the COL (2953%) biomaterials denoted significant statistical differences compared to the other biomaterials (1445%, 893%, and 1356% for COL:BG, COL:D.E., and COL:BS, respectively), confirming the effect of the incorporation of silica-based materials in the collagen matrix regarding the absorption capacity since a more cohesive structure seemed to have been formed (in coherence to the increase in mechanical properties above mentioned). The swelling of COL:BG, COL:D.E., and COL:BS achieved saturation after 5 min of soaking in PBS, while COL kept swelling with equilibrium and was attained only at about 20 min of incubation.

### 2.7. In Vitro Cellular Assays

#### 2.7.1. Biomaterials Cytotoxicity Assay

The eventual cytotoxicity of the developed biomaterials was assessed by quantitatively evaluating the metabolic activity of L929 cells cultured into them using the MTS assay. The obtained results (Figure 6) show that L929 cells exhibited an increase in metabolic activity after seeding on top of the biomaterials in all the developed structures for up to three days. This finding suggests the cytocompatibility of the used materials and the suitability of the processing methodology to fabricate the 3D structures without compromising the biological activity. The introduction of silica-based materials into the marine collagen matrix did not show any deleterious effect on cell metabolic activity. As previously reported, silica-based biomaterials are of special interest due to their capability to form chemical bonds with different living tissues [69]. When comparing the different silica-based materials, the results are quite similar in the first two days, but after three days of culture, BS stands out with a statistically significant higher metabolic activity, which may be related to the bioactive nature of the biogenic silica isolated from marine sponges. This is in agreement with the results reported in the literature, as biosilica from marine sponges has recently shown a favorable influence on the cell viability of the osteoblast precursor cell line (MC3T3-E1) and has been described to increase the hydroxyapatite formation in human osteoblast-like SaOS-2 cells.

Globally, silica-based materials have been demonstrated to have promising properties for biomedical applications [70,71]. Similarly, marine collagen scaffolds seeded with fibroblasts have been showing efficient cell growth and proliferation with importance for medical health applications [72,73,74]. Therefore, based on a strategy of marine by-product valorization, the combination of both materials seems to be a promising way to fabricate biomimetic and functional materials as herein proposed, with the use of the marine sponge silica revealing more promising results.

#### 2.7.2. Calcein Staining

Calcein is a cell-permeant dye that is transformed by the action of cytosolic esterases into green, fluorescent compounds in only metabolically active cells, which allows the identification of live cells. The cell-seeded biomaterials stained with calcein dye showed a strong increase in the green signal intensity from day 1 to day 7 (Figure 7). These qualitative calcein results are indicative that all conditions of biomaterials allowed the adhesion of L929 cells without significant macroscopic differences observed between them, in agreement with the results observed by others with collagen templates for the cell culture [75]. The increase in green signal intensity throughout the cell culture time, together with the previously mentioned increment of metabolic activity, suggests the proliferative state of the cells. These biomaterials seem to offer a structural environment for hosting fibroblast cells, as suggested by the high density in the green, fluorescent calcein cells that adhered to the biomaterials.

#### 2.7.3. Phallodin/DAPI Staining

Rhodamine-phalloidin (red color) and DAPI (blue color) staining evaluated cell distribution and cytoskeleton cell conformation and were, herein, used to evaluate the morphology of L929 cells cultured in the four different biomaterials for 7 days. DAPI stained the nucleus of the cells while rhodamine-phalloidin binding and stabilized filamentous actin (F-actin) occurred in the external structures of cells, and the obtained results (Figure 8) illustrate a uniform distribution of cells in the different scaffolds with no apparent differences between the different formulations. As could be observed 7 days after the cell seeding, the cells showed a more elongated shape in comparison with day 1, suggesting the potential of collagen for the cell culture. Similarly, in the literature, it has been reported that collagen isolated from *C. bathachus* fish increased the cell adhesion, proliferation, and migration of fibroblast [76]. In fact, the use of marine collagen in the biomedical field has been growing as a safe alternative compared to bovine collagen, which might be associated with the transmission of diseases [77] and allergenic effects. The in vitro cellular assays performed in this work support the utilization of collagen from codfish by-products, a low-cost raw material, for the production of marine biomaterials envisaging tissue engineering, given the observed good adhesion and proliferation of fibroblasts in the developed scaffolds. On the other hand, studies have been demonstrating the positive effect of silica-based materials on cell response [38,61,78]. The addition of diatom particles to silk fibroin improved the osteogenic properties of osteoblast-like cells compared with pure silk fibroin [60]. Le et al. also investigated the biological response of human mesenchymal stem cells (hMSCs) cultured on diatom-loaded fibroin sponges, with results confirming that osteogenic activity on the diatoms-loaded silk fibroin scaffolds was improved, with an increase in alkaline phosphatase activity (ALP) and early fibronectin and collagen type I formation observed [79]. Furthermore, diatom particles revealed no cytotoxic effect in concentrations up to 500 μg/mL [16]. However, some studies have reported that the concentration of silica influences the performance of cell behavior. A silica:collagen ratio ≤1 demonstrates favorable surface biocompatibility, while an additional silica concentration has a negative impact [41]. Recently, scaffolds printed with silica-doped hydroxyapatite ceramic to mimic the natural trabecular bone structure showed that there was no potential cytotoxicity for L929 cells [80]. Our results corroborated with the previous findings reported in the literature, reinforcing the non-cytotoxicity effect of silica-derived materials and suggesting the potential of biosilica for bone tissue engineering.

## 3. Materials and Methods

### 3.1. Raw-Materials

Frigoríficos da Ermida. Lda, a fish processing industry based in Gafanha da Nazaré, Portugal, kindly provided the Atlantic codfish (*Gadus morhua*) skins previously preserved in salt brine. These skins were by-products of the fish processing for food. These by-products were transported frozen to the laboratory facilities and stored at −20 °C until further use.

The silica-based materials were bioglass (BG), diatomaceous earth (D.E.), and biosilica (BS) retrieved from *Axinella infundibuliformis* marine sponge. Bioactive glass 45S5 bioglass^®^ (BG) was purchased from NOVABONE, Alachua, FL, USA, while Fossil Shell Flour^®^ diatomaceous earth (D.E.), from PERMA-GUARD^TM^, was kindly offered by Sollaris (Poland). Marine sponge *Axinella infundibuliformis* was collected by Prof. Hans Tore Rapp (University of Bergen, Norway) at Korsfjorden, Norway, preserved in absolute ethanol, and was kindly offered for further studies. Samples of marine sponge *A. infundibuliformis* were abundantly washed with ultrapure water to remove any exogenous materials and ethanol used for preservation. Biosilica was obtained after the calcination of the sponge samples at 800 °C for 6 h in a furnace (Termolab, BL1700, Águeda, Portugal).

### 3.2. Marine Collagen Extraction

The frozen codfish skins were thawed at a temperature between 4 and 7 °C. The collagen extraction was undertaken in a cold room (~4 °C) to avoid collagen denaturation, following the extraction steps as described in Figure 9.

The collagen extraction yield (% wet weight) was calculated using the following equation:$$\mathrm{Yield}\;\left(\%\right)=\frac{\mathrm{Weight}\;\mathrm{of}\;\mathrm{collagen}\;\left(g\right)\;}{\mathrm{Weight}\;\mathrm{of}\;\mathrm{wet}\;\mathrm{skins}\;\left(g\right)\;}\times \;100$$

### 3.3. Collagen and Silicas Characterization

#### 3.3.1. Sodium Dodecyl Sulfate-Polyacrylamide Gel Electrophoresis (SDS-PAGE)

SDS-PAGE is an electrophoresis technique used for the separation of proteins based on their different migration speeds in a gel, according to the differences in molecular weight. SDS-PAGE was performed using the SDS gel preparation kit (Sigma-Aldrich^®^, St. Louis, MO, USA) according to the manufacturer´s instructions. A commercial collagen solution from bovine skin type I (Sigma-Aldrich^®^) was prepared at 1 mg/mL as it is used to obtain the collagen standard profile. The extracted marine collagen was dissolved at 1 mg/mL in 0.5 M acetic acid solution, and the 1× protein assay dye reagent (Bio-Rad) was added to the bovine and codfish collagen solutions at a 1:1 (*v/v*) ratio. The mixtures were heated at 95 °C in a water bath for 15 min and centrifuged at 10.000× *g* for 1 min to remove the undissolved material. The supernatants were loaded (10 µL) to the prepared polyacrylamide gel with 3% of the stacking gel and 7.5% of the preparation gel and cast on the Mini Protean^®^ 3 Cell System (Bio-Rad, Hercules, CA, USA). The PageRuler™ Plus Prestained Protein Ladder (Thermo Scientific™, Vilnius, LT, USA) was added (4 µL) to estimate the molecular weights of the collagen samples. The samples were run at 90 V for 45 min until the frontline of the samples reached the lower part of the gel. After electrophoresis, the gel was stained with Coomassie Blue R 250 (Bio-Rad) for 30 min and rinsed with a destaining solution (70% distilled water, 20% methanol, and 10% acetic acid glacial) under gentle agitation until the protein bands were adequately visualized.

#### 3.3.2. FTIR Spectroscopy Analysis

Lyophilized collagen and bioglass, diatomaceous earth, and biosilica powders were analyzed by a Fourier Transformed Infrared (FTIR) spectroscopy. The samples were mixed with KBr (1:100) using a hand-operated press to form thin and transparent pellets for subsequent FTIR spectrum recording. Each pellet was analyzed using an IRPrestige 21 spectrometer (Shimadzu, Kyoto, Japan)) with a spectral resolution of 2 cm^−1^ and registered in the transmittance (%) mode, with each spectrum performing an average of 32 scans in the range of 800–4000 cm^−1^.

#### 3.3.3. CD Spectroscopy Analysis

Circular dichroism (CD) spectroscopy was used to assess the secondary structure of the collagen. The CD was performed using the Jasco J-1500 CD spectrometer (Tokyo, Japan) and a 2 mm quartz cuvette filled with 600 µL of 0.1 mg/mL collagen solution in 50 mM of acetic acid. The acquisition of the spectra was performed at 4 °C with continuous wavelength scans (in triplicate), from 180 to 240 nm, at a scanning speed rate of 50 nm/min.

#### 3.3.4. Amino Acids Analysis

The amino acid content of the collagen was determined using a Biochrome 30 (Biochrome Ltd., Cambridge, UK) amino acids analyzer. A total of 5 mg of the lyophilized collagen was completely hydrolyzed under acidic conditions (6N HCl) in vacuum-sealed glass tubes at 110 °C for 20 h. The hydrolysate was vaporized and the residue was dissolved in a sodium citrate buffer pH 2.2, and the amino acids were separated using a cation column. Subsequently, the column eluent was mixed with a ninhydrin reagent and was eluted at a high temperature. This mixture reacted with the amino acids forming colored compounds that were analyzed at 440 and 570 nm wavelengths. An internal standard of 10 nmol norleucine and 10% of the hydrolysate were used for quantitative analysis. Three independent measurements for each sample were performed for the quantification of the average amino acid contents.

The ratio of proline hydroxylation was calculated following the equation:$$\mathrm{Hydroxylation}\;\left(\%\right)=\frac{\;\mathrm{OHPro}\;\mathrm{content}}{\;\mathrm{pyrrolidine}\;\mathrm{amino}\;\mathrm{acid}\;\mathrm{content}}\times 100$$
where the pyrrolidine amino acid content corresponds to the sum of the hydroxyproline (OHPro) and proline (Pro) amino acids.

### 3.4. Production of Biomaterials: Collagen and Collagen-Silica-Based Scaffolds

The building blocks used for the fabrication of biomaterials were marine collagen (COL) isolated from codfish skins and silica-derived materials commercially available as the 45S5 Bioglass^®^ particulate (BG, NovaBone, USA), Diatomit-Pure Diatomaceous earth product (D.E., PERMA-GUARD^TM^, Poland), and biosilica isolated *A. infundibuliformis* sponge (BS). Four different biomaterial formulations were prepared by a freeze-drying methodology (Figure 10 and were designated as (1) COL, (2) COL:BG, (3) COL:D.E., and (4) COL:BS). Briefly, for the COL formulation, 1.5% (*w/v*) marine collagen was dissolved in 0.02 M acetic acid solution at 4 °C and was covalently crosslinked with 10% (*v/v*) of 120 mM 1-ethyl-3-(3-dimethylaminopropyl)-carbodiimide hydrochloride (EDC, Sigma Aldrich, UK) solution at a low temperature. To obtain the composite formulations (COL:BG, COL:D.E., and COL:BS), the respective silica-based material was combined with the collagen (a ratio of 64.5% and 35.5%, respectively) prior to crosslinking with EDC. The mixtures were homogenized by stirring, and air bubbles were removed by speed centrifugation at 4000 rpm for 5 min. All 4 formulations were transferred into 96 plate molds with 200 µL solution and were frozen at 20 °C for 3 h to crosslink with the solution. Afterward, the biomaterials were washed abundantly with ultrapure water (5 min × 3), and they were then frozen at −80 °C for 24 h, freeze-dried, and stored at room temperature until use.

### 3.5. Morphological Analyses

The image acquisition of freeze-dried 3D biomaterials was taken using the Stemi 2000-C stereomicroscope (ZEISS, Göttingen, Germany) attached to a 1.4-megapixel resolution digital camera (AxioCam ICC1, Göttingen, Germany).

#### 3.5.1. Scanning Electron Microscopy

The surface morphology of the biomaterials (COL, COL:BG, COL:D.E., and COL:BS), as well as the powder of BG, D.E., and BS, were analyzed through a Scanning Electron Microscope (SEM) (JSM-6010 LV. JEOL., Tokyo, Japan). Firstly, the samples were sputter-coated with an electrically conducted layer of platinum, and the outer structure of each sample was investigated. The micrographs were obtained at 10 kV in different magnifications.

#### 3.5.2. Microcomputed Tomography

The internal structures of the four different fabricated biomaterials were performed on a high-resolution µCT equipment Skycan 1272 (Bruker, Kontich, Belgium), using a pixel size of 7.5 µm. The X-ray source was set at 50 kV with a current of 200 μA. The tridimensional structure of the composites was acquired using a set of approximately 490 projections over a rotation range of 360° and a rotation step of 0.4°. Data sets were reconstructed using standardized cone-beam reconstruction NRecon^®^ software (v1.1.3), and the segmentation of layers into binary images was performed using a threshold of 25–255. The morphometric parameters, including porosity, pore size, interconnectivity, pore wall thickness, and the degree of anisotropy were calculated in triplicates per condition (COL, COL:BG, COL:D.E. and COL:BS) using CT Analyser® software (v1.15.4.0).

### 3.6. Energy Dispersive X-ray Spectroscopy

The characterization of chemical elements presented on the powder of BG, D.E., and BS and the surface of the biomaterials was assessed by energy dispersive X-ray spectroscopy (EDS) using the INCAx-Act. PentaFET Precision (Oxford Instruments, Concord, UK) was used at an energy of 10.0 keV coupled to SEM.

### 3.7. Mechanical Properties

The mechanical properties of the fabricated freeze-dried 3D structures were determined using universal mechanical testing equipment Instron 5543 (Norwood, OH, USA), with a 1 kN load cell in an unconfined compressive mode. Five different samples of each condition were compressed in uniaxial compression at a rate of 2 mm/min up to a strain level of approximately 80% of its height. Data were obtained using Bluehill V2.9.512 software, and the compressive modulus was determined from the slope of the initial linear region in the stress–strain curve.

### 3.8. Absorption Capacity

The absorption capacity of the biomaterials (COL, COL:BG, COL:D.E, and COL:BS) was investigated upon incubation in the phosphate-buffered saline solution (PBS, pH 7.4) for 1, 3, 5, 10, 15, 20, 30, and 60 min at room temperature (*n* = 5). After immersion for the determined time, the samples were blotted dry on filter paper to remove excess water, weighed, and the absorption ratio (%) of the biomaterials was calculated according to the following equation:$$\mathrm{Absorption}\;\mathrm{capacity}\;\left(\%\right)=\frac{\mathrm{wf}-\mathrm{wi}\;}{\mathrm{wi}}\times 100$$
where wf represents the weight of the wet biomaterial and wi is the initial dry weight of the biomaterial.

### 3.9. In Vitro Biological Assays

#### 3.9.1. Cell Culture

A well-established immortalized mouse fibroblast cell line (L929 cells) purchased from the European Collection of Cell Cultures (ECACC, UK) was grown in Dulbecco’s modified eagle medium (DMEM, Sigma) supplemented with 10% Fetal bovine serum (FBS, Biochrom) and 1% antibiotic/antimycotic solution (Invitrogen) at pH 7.4 and incubated at 37 °C in a humidified atmosphere with 5% CO_2_. The cell dissociation from the confluent culture flasks used a chemical method with trypsin-EDTA (0.05%. ThermoFisher scientific).

#### 3.9.2. Cell Viability and Cytotoxicity Assay on Scaffold Composites

Aiming to evaluate any cytotoxic effect resulting from the used materials or the processing methodology, direct contact cytotoxicity assays were performed by using an L929 cell line. The cells were expanded in DMEM until 80% of confluence for further use. The biomaterials were disinfected in 70% ethanol solution for 1 h, washed extensively with sterile phosphate-buffered saline (PBS), and were then washed with DMEM culture medium. The cells were seeded onto the biomaterials with a final density of 2.5 × 10^4^ cells per biomaterial by using a droplet (100 µL) approach. To increase the seeding efficiency, the drop was maintained in contact with the scaffolds for 4 h (37 °C, 5% CO_2_), and a final volume of the culture medium (500 µL) was added afterward. The metabolic activity of the cells cultured on the scaffold composites was quantified by using a colorimetric MTS assay. MTS is a yellow water-soluble tetrazolium compound, which is bio-reduced by living cells with the production of a purple formazan product. The amount of purple product generated is directly proportional to the number of metabolically active cells. MTS was prepared by adding the MTS solution (CellTiter 96^®^ Aqueous One Solution Cell Proliferation Assay (MTS), Promega) following the manufacturer’s specifications. Briefly, a serum-free culture medium without phenol red was used for the reaction of MTS (5:1, medium: MTS reagent). For all assays, a blank control (only MTS medium) and positive controls (L929 cell line seeded onto the bottom of 48 well plates) were used. The reaction was incubated for 4 h at 37 °C in a 5% CO_2_ atmosphere. The MTS color exchange was quantified by measuring the absorbance at 490 nm by using a microplate reader. The absorbance of the results was corrected by removing the blank value. The cytotoxicity test was evaluated for 1, 2, and 3 days of the cell culture, and each experimental condition was tested in triplicates, and three independent assays were performed. The results were analyzed as the mean ± SD.

#### 3.9.3. Calcein Staining

To qualitatively assess the cell viability in the different produced biomaterials, L929 cells were seeded in the scaffolds, as previously mentioned, at a density of 2 × 10^5^ cells per scaffold and incubated at 37 °C in a 5% CO_2_ atmosphere. The culture medium was changed every 2–3 days. After 1, 3, and 7 days of cell culture, the cells in the biomaterials were stained using a green fluorescence calcein AM cell-permeant dye (ThermoFisher Scientific) at a ratio of 1:600 and incubated in the dark for 5 min at 37 °C and 5% CO_2_. Afterward, the biomaterials were abundantly washed in sterile PBS and visualized using a transmitted and reflected light microscope (Axio Imager Z1m, Zeiss). Three samples of each experimental condition were analyzed. Additionally, a control (biomaterial without seeded cells) was used to discard material autofluorescence.

#### 3.9.4. Phalloidin/DAPI Staining

The biomaterials were seeded with L929 cells at a density of 2 × 10^5^ cells per scaffold and incubated at 37 °C in a humidified atmosphere with 5% CO_2_. After 1, 4, and 7 days of culture, the composites were washed with PBS and fixed with 10% formalin overnight. After formalin removal, the scaffolds were washed with PBS, and a 0.1% (v:v) triton solution was added for 5 min at room temperature. The biomaterials were incubated with a solution of phalloidin–tetramethylrhodamine B isothiocyanate (1:100)/4.6-Diamidino-2-phenyindole dilactate (DAPI. Sigma) (1:5000) diluted in 1× PBS for 45 min and protected from light. Finally, the biomaterials were washed in 1% PBS and were visualized under fluorescence microscopy (Axio Imager Z1m, Zeiss) to evaluate the morphology and cytoskeletal organization of the L929 fibroblast cells on the biomaterials.

### 3.10. Statistical Analysis

The statistical differences between the experimental conditions were analyzed using GraphPad Prism 8 software following one-way or two-way ANOVA and the Tukey post hoc test. If the data did not meet the normality, the mean differences between the all-different 3D biomaterials were compared using a nonparametric test (Kruskal–Wallis test) followed by Dunns’ post hoc test with a minimum confidence level of 0.05 for the determination of statistical significance. All the performed tests were conducted at least in triplicate (*n* = 3), and the obtained data points are presented as the mean ± standard deviation (Mean ± SD).

## 4. Conclusions

The codfish skin by-product generated during the industrial processing of codfish could be used as a raw material for the production of collagen with biochemical properties resembling type I collagen and is, herein, proposed as an alternative to mammalian collagen for biomedical purposes. In addition, biosilica could be retrieved from *A. infundibuliformis* marine sponge to obtain needle-like microparticles. These materials were used to produce scaffolds using freeze-drying and compared with others produced with alternative silica-based materials, namely, bioactive glass and diatomaceous earth. The developed collagen and composite scaffolds showed high porosity (P > 85%) and interconnected (I > 70%) pores with diameters between 138 and 315 µm without significant differences between the formulations. The addition of silica-based components promoted an increase in mechanical properties, particularly when using bioglass, with all structures showing a very high (>1000%) water uptake, particularly for the pure collagen formulation. Regarding in vitro biological performance as templates for 3D cell cultures, in general, all of the structures proved to be non-cytotoxic materials (MTS test), and high cellular density was visible by staining with calcein dye, with adhered cells distributed by the whole structures, as detected upon phalloidin/DAPI staining. Moreover, the highest cell viability (L929 fibroblast cells) was exhibited in the composite scaffold comprising biosilica derived from the marine sponge. From the obtained results, these composites may be useful biomaterials for the engineering of mineralized tissues, namely bone, and contribute to an increasing awareness of the potential of marine origin materials for biomedical application. In addition, it also adds to the opportunities offered by the sustainable use of marine resources and derived by-products.

## Figures and Tables

**Figure 1 marinedrugs-20-00718-f001:**
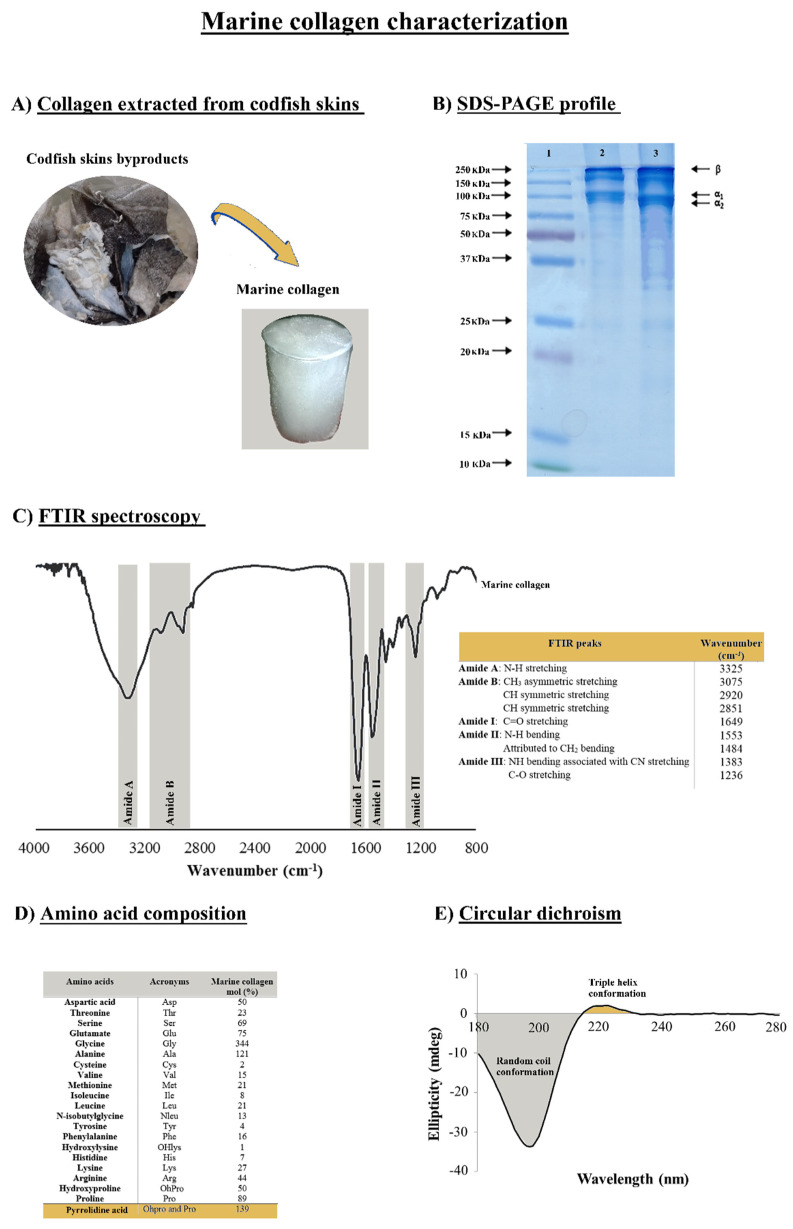
Marine collagen isolated from codfish skins and its characterization. (**A**) Collagen extracted from fish skins. (**B**) SDS-PAGE gel electrophoresis of type I collagen wherein represents 1—PageRuler^®^ Plus Prestained protein ladder, 2—collagen standard from bovine skin (Sigma) and 3—collagen profile (1%) isolated from codfish. (**C**) FTIR spectrum and values of the amide peaks. (**D**) Amino acid composition of codfish collagen (expressed as residues/1000 residues). (**E**) Circular dichroism spectrum of the collagen.

**Figure 2 marinedrugs-20-00718-f002:**
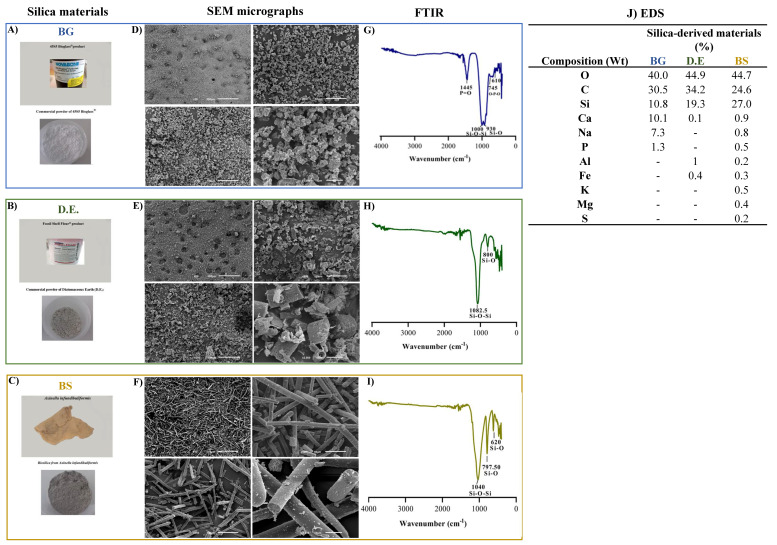
Silica−derived materials with characterization of (**A**) Bioglass^®^ 45S5 (BG); (**B**) Diatomaceous earth (D.E.); and (**C**) Biosilica (BS) raw materials. (**D**–**F**) SEM micrographs of the BG, D.E., and BS powders, respectively, at 50×, 250×, 500×, and 2000× magnifications. (**G**–**I**) FTIR spectra of the BG, D.E., and BS materials, respectively, and (**J**) Chemical composition of the silica-derived materials by EDS analysis.

**Figure 3 marinedrugs-20-00718-f003:**
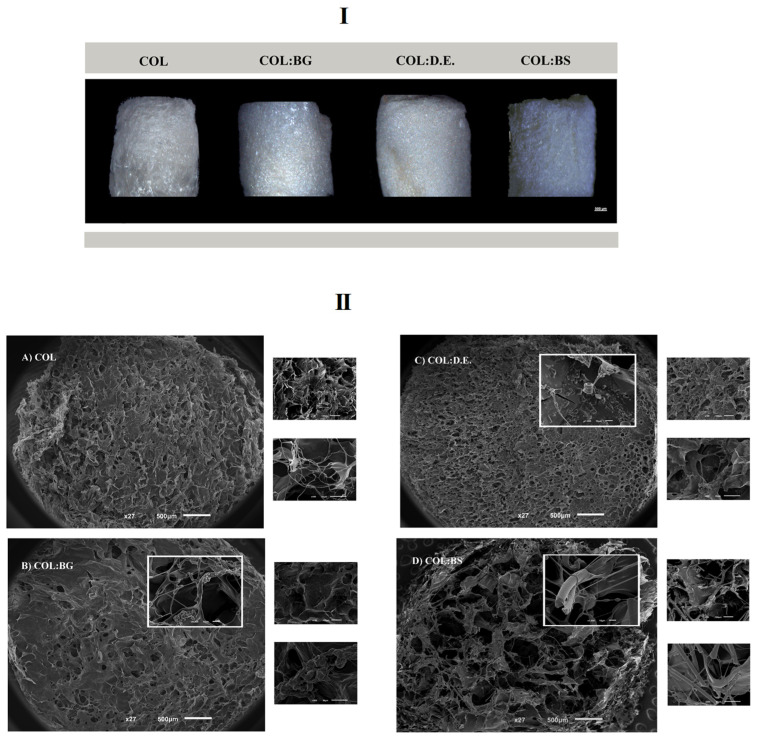
Morphological features of the four developed 3D scaffold formulations: COL, COL:BG, COL:D.E., and COL:BS. (**I**) Photographs obtained with stereomicroscope. (**II**) SEM micrographs obtained at different magnifications (27×, 150×, 500× and 1000×).

**Figure 4 marinedrugs-20-00718-f004:**
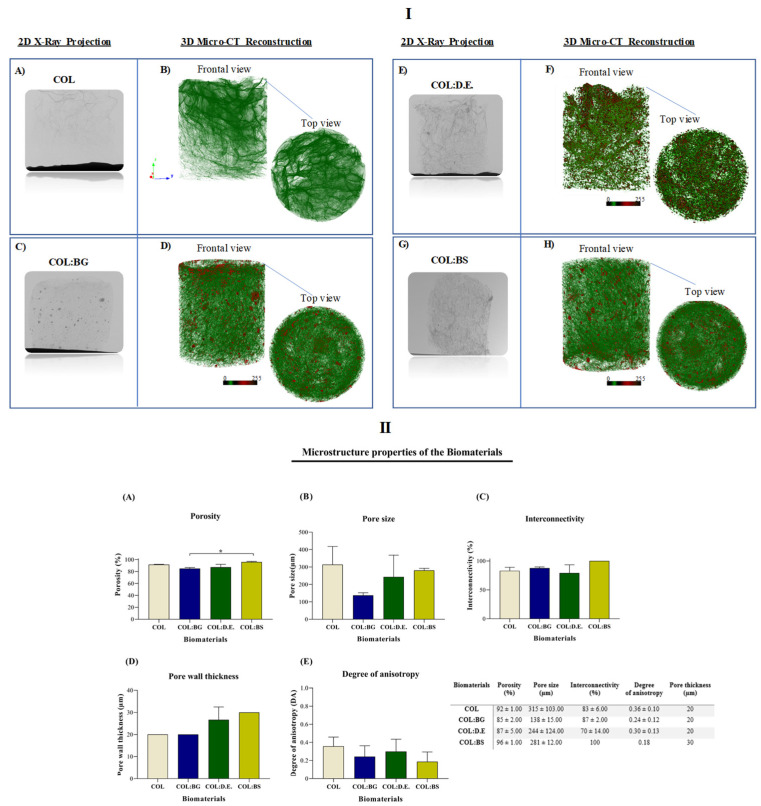
Micro-CT analysis of the developed COL, COL:BG, COL:D.E., and COL:BS scaffolds, including: (**I**) Two-dimensional X-ray projections (**A**,**C**,**E**,**G**) and tridimensional reconstruction images of the frontal and top views (**B**,**D**,**F**,**H**) for the four the biomaterials; (**II**) Morphometric properties, namely, (**A**) Porosity (%), (**B**) Pore size (µm), (**C**) Interconnectivity property (%), (**D**) Pore wall thickness (µm) and (**E**) Degree of anisotropy. Data are expressed as mean ± SD (*n* = 3) and the asterisks indicate statistically significant differences (*p* < 0.05).

**Figure 5 marinedrugs-20-00718-f005:**
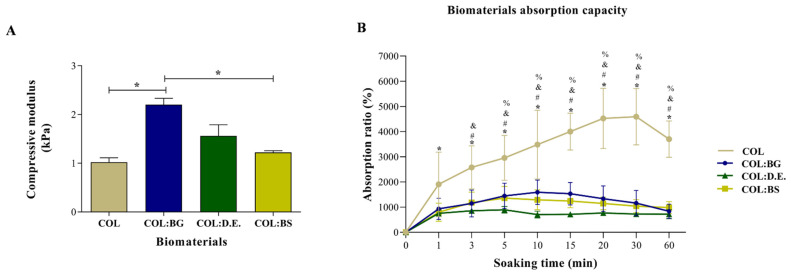
Mechanical and swelling properties of biomaterials. (**A**) Compressive modulus of COL, COL:BG, COL:D.E., and COL:BS biomaterials. The statistically significant differences were determined using one-way ANOVA with post hoc Tukey’s test (*: *p* < 0.05). Values are represented as mean ± SD. (**B**) Changes in absorption capacity of the developed biomaterials after incubation in PBS. Symbols denote statistically significant differences (*p* < 0.05) in comparison to: (*) COL 0 min, (#) COL vs. COL:BG, (and) COL vs. COL:D.E., and (%) COL vs. COL:BS.

**Figure 6 marinedrugs-20-00718-f006:**
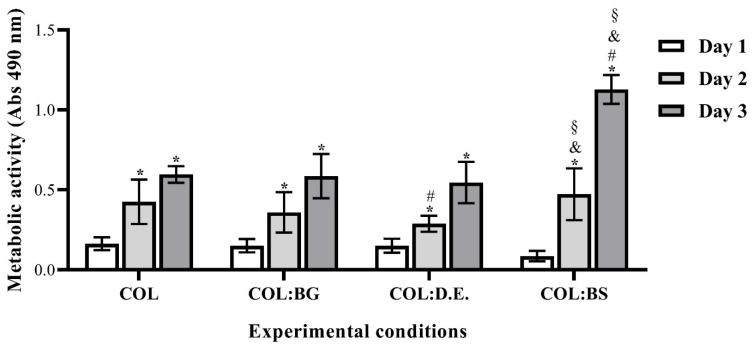
Metabolic activity of L929 cells seeded on the top of each biomaterial (COL, COL:BG and COL:D.E., and COL:BS) and cultured for 1, 2, and 3 days, as determined by using MTS assay. Data are expressed as mean ± SD of three independent experiments with triplicates. Symbols denote statistically significant differences (*p* < 0.05) in comparison to: COL (#), COL:BG (and), COL:D.E (§) at the same time point on day 1 vs. 2 or day 1 vs. day 3 into the same condition (*).

**Figure 7 marinedrugs-20-00718-f007:**
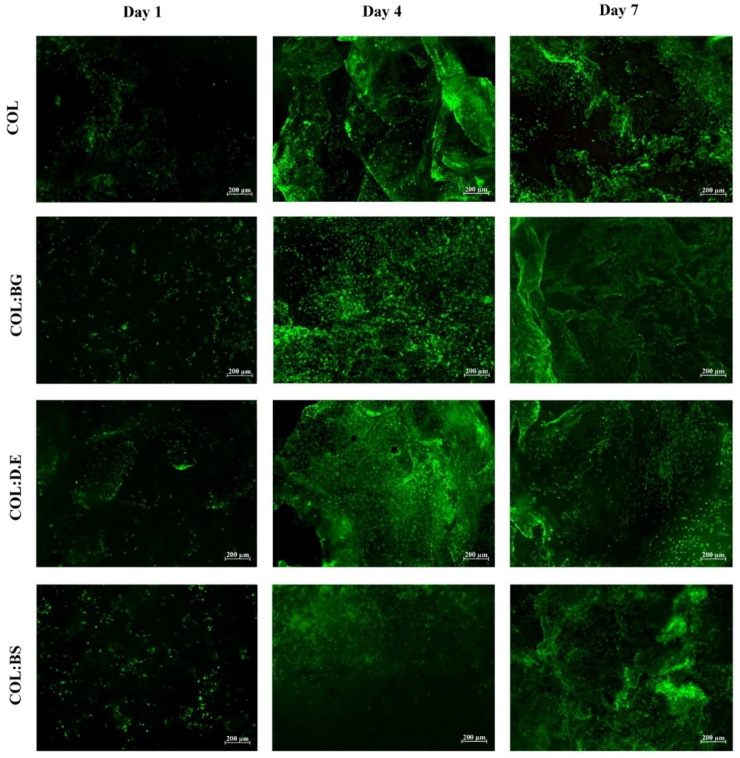
Fluorescence microscopy images of calcein-AM staining of L929 fibroblast cell line that adhered on the developed biomaterials (COL, COL:BG, COL:D.E., and COL:BS) after 1, 4, and 7 days of cell culture.

**Figure 8 marinedrugs-20-00718-f008:**
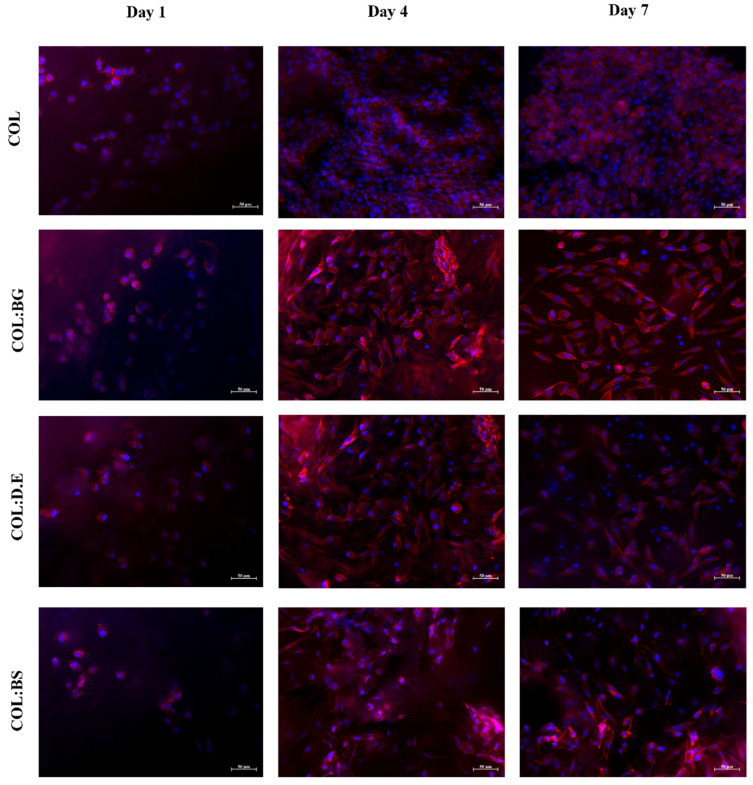
Fluorescence microscopy images of rhodamine-phalloidin and DAPI stained L929 cells grown in the produced composite biomaterials (COL, COL:BG, COL:D.E, and COL:BS) for 1, 4, and 7 days. Rhodamine phalloidin (red) and DAPI (blue) with stained F-actin and nuclei of L929 cells, respectively. The cells shown are representative of the data (*n* = 3).

**Figure 9 marinedrugs-20-00718-f009:**
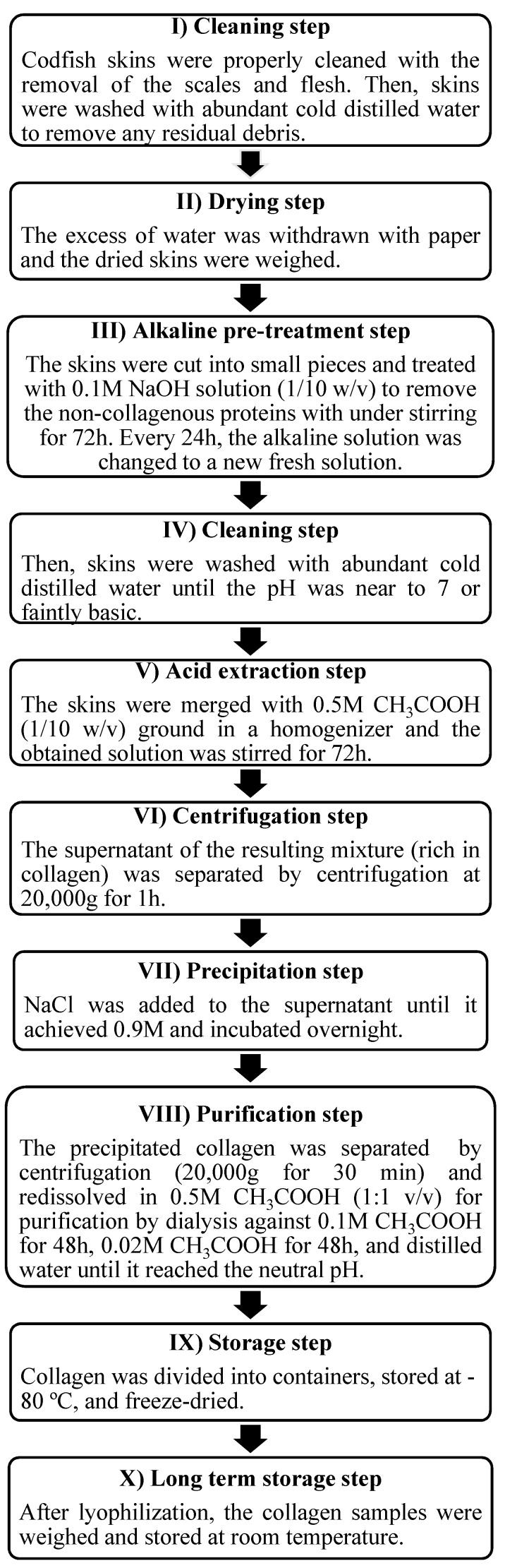
Scheme of the methodology to isolate collagen from *Gadus morhua* codfish skins.

**Figure 10 marinedrugs-20-00718-f010:**
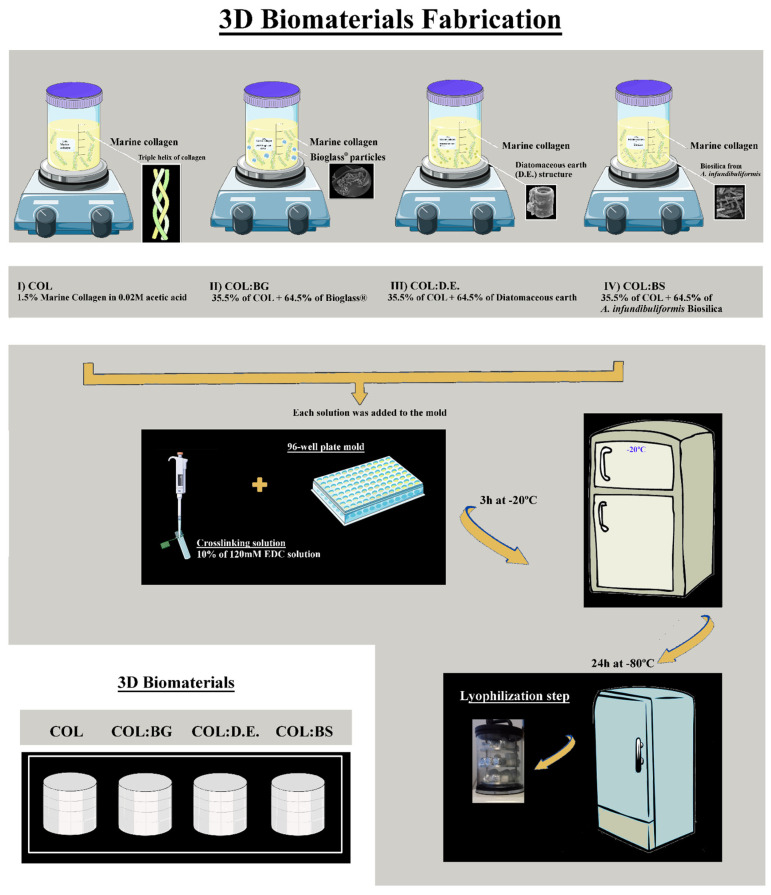
Scheme of the methodology used for the fabrication of 4 different formulations of marine collagen and composite scaffolds, namely, COL, COL:BG, COL:D.E., and COL:BS.

**Table 1 marinedrugs-20-00718-t001:** Elemental chemical composition weight (%) of the 3D biomaterials obtained by EDS analysis.

3D Biomaterials	C	O	Cl	S	Na	Ca	N	Si	K
**COL**	58.8	25	1.7	0.7	-	-	13.7	-	-
**COL:BG**	44.8	21.3	1.2	0.6	0.2	0.4	30.7	0.8	-
**COL:D.E.**	43.5	21.8	2.9	0.4	0.1	0.7	26	4.7	-
**COL:BS**	53	21.6	3.2	0.4	-	-	17.4	4.4	0.1

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
