# Peer review of "3D Biocomposites Comprising Marine Collagen and Silica-Based Materials Inspired on the Composition of Marine Sponge Skeletons Envisaging Bone Tissue Regeneration"

_marinedrugs, 2022, doi:10.3390/md20110718_

Round 1
Reviewer 1 Report
Review of the manuscript entitled ‘3D biocomposites comprising marine collagen and silica-based materials inspired on the composition of marine sponge skeletons envisaging bone tissue regeneration’ by E. Martins et al.
The manuscript presents the preparation and characterization of hybrid organic/inorganic composites made of fish-origin collagen and three silica-based materials, namely bioactive glass, Diatomaceous Earth and biosilica isolated from sponges. The topic is evidently very actual as it relates to the biobased and biocompatible materials for medical applications – bone regeneration. Invented materials are probably applicable in this field however some issue should be clarified before manuscript publication. Manuscript is clearly written and the well-organized. I recommend it for publication in Marine Drugs after minor revision.
List of remarks and questions:
Line 239: The Authors mentioned about the uniform porous structure. What uniformity are the authors writing about here? On what scale? The pore size results presented in Figure 4 and described in lines 264-267 show a very large dispersion of the pore size, which suggests structure non-uniformity.
Lines 318-330: The Authors discussed here collagen reinforcement by inorganic particles. The reinforcement effect is rather low, and the obtained modules raise doubts as to whether the produced materials can be used in practice. Especially that only dry materials were tested. How do the composites behave in terms of mechanical properties in the hydrated state?
Author Response
Review of the manuscript entitled ‘3D biocomposites comprising marine collagen and silica-based materials inspired on the composition of marine sponge skeletons envisaging bone tissue regeneration’ by E. Martins et al.
The manuscript presents the preparation and characterization of hybrid organic/inorganic composites made of fish-origin collagen and three silica-based materials, namely bioactive glass, Diatomaceous Earth and biosilica isolated from sponges. The topic is evidently very actual as it relates to the biobased and biocompatible materials for medical applications – bone regeneration. Invented materials are probably applicable in this field however some issue should be clarified before manuscript publication. Manuscript is clearly written and the well-organized. I recommend it for publication in Marine Drugs after minor revision.
List of remarks and questions:
Line 239: The Authors mentioned about the uniform porous structure. What uniformity are the authors writing about here? On what scale? The pore size results presented in Figure 4 and described in lines 264-267 show a very large dispersion of the pore size, which suggests structure non-uniformity.
Reply: We thank the reviewer for your comments. We change the sentence (line 239) to better describe the results of morphological analysis. Despite SEM micrographs showed, apparently, an uniform pore size in each formulation, the quantitative analysis performed by microCT (Figure 4) have shown that the developed structures had a porous structure with different pore sizes between all biomaterials and in some cases with a considerable dispersion.
Lines 318-330: The Authors discussed here collagen reinforcement by inorganic particles. The reinforcement effect is rather low, and the obtained modules raise doubts as to whether the produced materials can be used in practice. Especially that only dry materials were tested. How do the composites behave in terms of mechanical properties in the hydrated state?
Reply: Thank you for your comment, dear reviewer. Our results demonstrated differences significantly between the COL biomaterials and the other COL-derived silica materials. However, we mentioned that it was a light reinforcement of the compressive modulated compared to the COL biomaterials. In the dry state, our biomaterials showed a low compressive modulus compared to the trabecular and compact bone, thus being a possibility for non-load bearing applications, and eventually requiring additional reinforcing strategies. Considering the published literature on collagen-based biomaterials, we hypothesized that the developed biomaterials would have a lower compressive modulus in a hydrated state than in a dry state, with the differences between COL biomaterials and COL-silica biomaterials being lower or even non-existent.
Reviewer 2 Report
There is no doubt that the manuscript entitled " 3D biocomposites comprising marine collagen and silica-based materials inspired on the composition of marine sponge skeletons envisaging bone tissue regeneratio" by Eva Martins, Gabriela S. Diogo , Ricardo Pires , Rui L. Reis , and Tiago H. Silva contains a lot of interesting results and it is worth to publish it in Marine Drugs. The authors show the possibility of using of marine origin materials for biomedical applications, particularly as composite scaffolds for bone regeneration. They used various methods of analysis and the results are generally well documented and clearly presented.The main problem which should be corrected before publication is related to the analysis of mechanical properties of investigated materials. Firstly, the reviewer supposes only that the stress in compression tests was determined without considering porosity. If so, the authors should mention this fact explicitly in "Materials and Methods" section. If this is not a case, the authors should mention in Methods and Materials section porosity incorporation in stress determination. Secondly, the only parameter determined from compression tests is Young modulus but showing its values for investigated materials in Fig. 5A, the authors used the term "Compressive strength" on Y-axis. The same term is used in line 329. The reviewer draws the authors' attention that the term "compressive modulus" and "compressive strength" are not equivalent. Shortly, the authors should change the the term "compressive strength" to "compressive modulus" being adequate to what was indeed measured by them. The last point is related to the Abstract in which the authors used materials abbreviations few rows earlier (rows 19, 20) before their full names accompanied by abbreviations. The reviewer can accept this fact provided that the authors will complete in row 24 the full name of biosilica with its abbreviation.
Author Response
There is no doubt that the manuscript entitled " 3D biocomposites comprising marine collagen and silica-based materials inspired on the composition of marine sponge skeletons envisaging bone tissue regeneratio" by Eva Martins, Gabriela S. Diogo , Ricardo Pires , Rui L. Reis , and Tiago H. Silva contains a lot of interesting results and it is worth to publish it in Marine Drugs. The authors show the possibility of using of marine origin materials for biomedical applications, particularly as composite scaffolds for bone regeneration. They used various methods of analysis and the results are generally well documented and clearly presented. The main problem which should be corrected before publication is related to the analysis of mechanical properties of investigated materials. Firstly, the reviewer supposes only that the stress in compression tests was determined without considering porosity. If so, the authors should mention this fact explicitly in "Materials and Methods" section. If this is not a case, the authors should mention in Methods and Materials section porosity incorporation in stress determination. Secondly, the only parameter determined from compression tests is Young modulus but showing its values for investigated materials in Fig. 5A, the authors used the term "Compressive strength" on Y-axis. The same term is used in line 329. The reviewer draws the authors' attention that the term "compressive modulus" and "compressive strength" are not equivalent. Shortly, the authors should change the the term "compressive strength" to "compressive modulus" being adequate to what was indeed measured by them. The last point is related to the Abstract in which the authors used materials abbreviations a few rows earlier (rows 19, 20) before their full names were accompanied by abbreviations. The reviewer can accept this fact provided that the authors will complete in row 24 the full name of biosilica with its abbreviation.
Reply: Thank you for your comment, dear reviewer. The mechanical properties of the biomaterials were measured using a universal mechanical equipment and compressive modulus results were determined by the slope of the initial linear region in the stress-strain curve, with the comparison between different biomaterial formulations not taking into consideration differences in porosity. In this regard, we agree with you that “compressive modulus” should be used. The manuscript was revised accordingly.
The abbreviation of biosilica was added in line 24, as the reviewer suggested.